# Research on Angle and Stiffness Cooperative Tracking Control of VSJ of Space Manipulator Based on LESO and NSFAR Control

**Xin Ye** , **Jia-Cai Hong and Zheng-Hong Dong** *

Space Engineering University, No.1 Bayi Road, Beijing 101416, China
* Correspondence: dzh.bj@163.com; Tel.: +86-138-1175-7848

**Abstract:** With the increase in on-orbit maintenance and support requirements, the application of space manipulator is becoming more promising. However, how to control the vibration generated by the space manipulator has been a difficult problem to be solved. The advent of variable stiffness joint (VSJ) has brought about a dawn in solving this problem. But how to achieve coordinated control of joint angle and stiffness is still a problem to be solved, especially when considering system model parameter uncertainty, unknown disturbance and control input saturation. In order to realize the controllable attenuation of the vibration of the space flexible manipulator based on the variable stiffness joint, the dynamic model of the variable stiffness joint was constructed. Then the linear transformation and feedback linearization method are used to transform its complex nonlinear dynamic model system into a pseudo-linear system containing aggregate disturbance and input saturation constraints. This paper constructs a linear extended state observer (LESO) for estimating the state of unknown systems in pseudo-linear systems. Based on the idea of state feedback control, a Neural State Feedback Adaptive Robust (NSFAR) control is constructed by using Radial Basis Function Neural Network. The adaptive input saturation compensation control law is also designed by using Radial Basis Function Neural Network to deal with the input saturation compensation problem. The ultimate uniform bounded stability of the constructed system is proved by the stability analysis based on Lyapunov function. Finally, the effectiveness and superiority of the constructed tracking algorithm are verified by compared simulation and semi-physical experiment.

**Keywords:** space manipulator; variable stiffness joint; feedback linearization; cooperative control algorithms for joint angle and stiffness; input saturation compensation; linear extended state observer; RBF Neural Network; state feedback control

## 1. Introduction

Nowadays, with the increasing frequency of space activities, the impact of spacecraft being hit by space debris has occurred. A large amount of space debris has already seriously threatened the safety of on-orbit spacecraft. The large number of large-scale space debris may change the attitude and orbit of spacecraft and even cause the spacecraft to be completely destroyed. For example, the large-capacity communication satellite named IS-29e, manufactured by Boeing, unfortunately exploded in space more than 35 million kilometers from the Earth in April, disintegrating into hundreds of fragments which are likely to collide with other spacecraft on the corresponding track during the offset [1].

In order to solve such problems, the development of space debris removal technology has become particularly urgent. Among the many active space removal technologies, the technology of space manipulator removal in orbit has received extensive attention. The ETS-VII of Japan, the Orbital Express of the United States, and the SY-7 of China have successively conducted verification experiments on this technology in space, and are still intensifying their research [2,3].

Related studies [4–6] have shown that when using the traditional multi-rigid manipulator to carry out space debris removal tasks, collisions from non-cooperative target contacts such as space debris may lead to large pulse momentum, which may cause problems such as space manipulator tumbling. The space manipulator equipped with a flexible mechanism can better achieve the collision force buffering and unloading during the contact with the non-cooperative target. It can solve the problem of energy shock and disturbance caused by satellites contact, make the contact process softer, reduce the various risks brought by the current rigid arm contact, and expand the practicality of space operation. At present, there is no real flexible manipulator to test or apply on orbit, but the research on space flexible manipulators has been extensive. Deshan et al. [7] established a dynamic model of the multi-link multi-DOF flexible manipulator and studied the vibration responses of the tip under different elastic modulus, damping and joint stiffness. Dong et al. [8] conducted a comprehensive study on the dynamics, kinematics modeling and rigid-flexible composite control of space manipulators based on soft contacts, and conducted comparative experiments using experimental data from existing the space manipulator. The above models are only applicable to the mechanical arm system with large joint stiffness, and the stiffness and damping of the joint are linear. The effects of nonlinear factors such as gap and friction are not considered. In order to meet the accuracy requirements of space work, it is necessary to establish a more accurate dynamic model of the manipulator for response characteristics analysis and controller design. OWAIS et al. [9] derived a Lagrangian-based dynamic model which lies in consideration of both viscous damping and gravity and proposed two Control algorithms based on Linear Quadratic Regulator (LQR) and nonlinear backstepping, respectively. Guo et al. [10] established the dynamic model in the presence of parametric uncertainties, unknown bounded friction torques, unknown bounded external disturbance, and input saturation constraints, by using the coordinate transformations and the static state feedback linearization, and designed a robust tracking controller by a combination of a disturbance observer, sliding mode control, and an adaptive input saturation compensation law.

We can find in this study that the inevitable vibration problem of the flexible manipulator makes it difficult to accurately control the actuator of the flexible arm. This situation is particularly serious in space, because the atmosphere is thin, and once the vibration is excited, it will be difficult to attenuate itself.

The appearance of variable stiffness joint (VSJ) solves this problem very well. When a space manipulator with variable stiffness joints comes into contact with the target, the joint can flex flexibly within a certain range to achieve the purpose of buffering the impact momentum while absorbing and storing the energy of the target. So that the base would not be overturned by a large disturbance, and the energy absorbed is stored in the elastic element of the variable stiffness joint, and can be released with control in the next step to maintain the entire space robot-target complex stable after contact [11].

The variable stiffness principle and configuration of the existing VSJ are numerous [12]. The related research work of VSJ is meanly on the following three types. Including: The VSJ whose joint stiffness is varied through the combination of two antagonistic serial elastic actuators (SEAs) controlled by two separate motors [13,14]. Other realizations for stiffness altering are achieved through the principle of lever mechanism [14–18], or else adjusting the preload of the linear spring by a nonlinear connector between the output link and the spring element [18–21]. VSJ has many advantages, mainly in the passive adaptability, inherent flexibility, and the ability to adapt to the task needs to adjust the joint output stiffness.

With the development of VSJ configuration research, research on the tracking control algorithm of VSJ has gradually emerged. Due to the different VSJ variable stiffness mechanisms, these studies are carried out for different structural types of VSJ or VSJ-driven multi-degree-of-freedom robot systems, all with different research objectives. It can be divided into separate tracking control for joint output angle and joint output stiffness of VSJ, and cooperative tracking control for joint output angle and joint output stiffness of VSJ. In terms of stiffness tracking control, the existing VSJ joint stiffness tracking control methods can be divided into two types: one is based on the VSJ joint stiffness mathematical

model to obtain the stiffness tracking control torque, and the other is to use the stiffness estimator to estimate the joint output stiffness.

PID or PD control algorithms are often used for tracking control of VSJ. Sun et al. used two PID controllers [22] or two PD controllers with feedforward compensation [16] to achieve tracking control of the output link angular position and joint output stiffness of SVSA.

Because of the nonlinearity of the stiffness adjustment of VSJ, the dynamic system is more complicated to construct. The feedback linearization method can transform the nonlinear system into a linear system while maintaining high precision. Therefore, the method is much more prominent is tracking control research of VSJ. Grebenstein et al. [23] designed a VSJ sliding mode tracking controller with an integral term based on feedback linearization, which uses the integral term to eliminate the impact of system model uncertainty, friction and unknown disturbance to the VSJ position and stiffness tracking control. Gabriele et al. [24] designed a recursive numerical algorithm based on the Newton–Eulerian dynamic equation, and a variant of the algorithm can be used to implement the feedback linearization control law to achieve the desired output link angular position trajectory and precise tracking control of joint output stiffness. In order to control the position and stiffness of bidirectional antagonistic drives independently, Kosta et al. [25] decoupled the systems in two linear single-input-single-output subsystems, position subsystem and stiffness subsystem, by feedback linearization. Sun [26] introduced the feedback linearization method into the walking control of the variable stiffness ankle joint and obtains a good control effect.

In addition, there are other control algorithms that are also used for tracking control of the VSJ's link output angle and joint output stiffness. For the tracking control problem of multi-degree-of-freedom robot based on VSJ, Petit et al. [27] designed the backstepping tracking controller, and used the instruction filter to deal with the state measurement noise and high-order state derivatives. And the effectiveness of the designed backstep controller is shown through the simulation and test based on BAVS-Joint and DLR hand arm system. Zhang et al. [28] designed a quasi-finite time tracking control law for the output link angular position tracking of an antagonistic VSJ based on wire rope transmission. The control law can achieve fast state response, high tracking accuracy, and good interference suppression performance. They [29] also designed a dynamic surface tracking control law based on forwarding technology to apply to the asymptotic tracking control of the output link angular position of the wire rope transmission antagonist type VSJ with non-matching disturbance.

There are not many related literatures on the robust control of VSJ, but there are different control objectives for different types of VSJ. For example, Branko Z. Lukiü et al. [30] introduced the feedforward neural network into the joint position and joint stiffness control of VSJ. The simulation results show that the feedforward neural network is significantly better than the open loop control, which significantly improves the trajectory tracking accuracy of joint stiffness. Huh and Bien [31] proposed a neuro-sliding mode approach based on model reference adaptive control (MRAC). The proposed MRAC control structure induces the VSJ to follow its nominal dynamics with help of sliding mode control efforts. The sliding gain, implemented by a simple neural network (NN), is adaptively updated based on the Lyapunov criterion. The simulation results show that this algorithm has advantages over the traditional PID control algorithm. Aiming at the tracking control problem of SVSA link output angle and joint output stiffness, Guo et al. [32] designed a neural network adaptive control algorithm based on feedback linearization. The simulation shows that the designed controller can cope with system model uncertainty and achieve link output angle and joint output stiffness tracking control. But it does not consider controlling input saturation constraints and controlling input saturation compensation issues during the design process. Psomopoulou et al. [33] designed a state feedback controller that achieves the desired tracking performance and is robust against internal and external disturbances in the motor. However, it does not consider the influence of other parameters of the system dynamics model such as inertia parameter and damping coefficient, on the tracking performance in the simulation, and the controller needs to consider the feasibility between the actuation characteristics of the actuator and the expected tracking performance of the closed-loop system. Zhang et al. [34] proposed an adaptive neural

network control scheme based on high-dimensional integral Lyapunov function, in order to achieve the desired output link angular position tracking performance, Radical Basis Function Neural Network (RBFNN) is used to approximate the unknown nonlinear function in the control law. Simulation and experiments show the effectiveness of the control method.

For the joint output stiffness control problem of VSJ, the nonlinear time-varying joint stiffness cannot be directly measured in real time, and the joint output stiffness derived from the VSJ mathematical model is prone to error. Therefore, Liu et al. [35] proposed to use the neural network observer control system design to solve the above unmeasurable problem. An observer based on RBFNN is used to estimate state variables of the normal system, with a controller based on dynamic surface control method for a single link flexible joint manipulator whose model is unknown. The unknown model of the manipulator is constructed by RBFNN. The effectiveness of the proposed controller is proved by the simulation. Jia [36] proposed a controller based on dynamic surface control and observer by using motor state feedback for trajectory tracking of flexible joint robot with uncertain link dynamic model. The simulation results show that the designed controller has a good trajectory tracking effect, which effectively suppresses the residual vibration of the flexible joint robot.

Although there are many studies on the method of VSJ's link output angle and joint output stiffness tracking control, there are still some problems. For example, the nonlinearity of the system dynamics model is high. The coupling between the link output angle tracking and the joint output stiffness tracking is strong. And there may be system dynamics model parameter uncertainty, unknown disturbance and control input saturation constraints. Therefore, it is necessary to study the robust control algorithm for joint angle and stiffness tracking of VSJ. The linear extended state observer (LESO) can effectively deal with the disturbance in the system. At present, there are few related studies on the VSJ's link output angle and joint output stiffness tracking control using this disturbance processing method [37]. Therefore, it is necessary to further study the cooperative tracking control method of the VSJ's link output angle and joint output stiffness based on this method. At the same time, it is found that in the tracking control of VSJ, the saturation of the control input amplitude may occur. If not considered in the design of the controller, it may affect the tracking performance or even the stability of the system. Therefore, it is necessary to consider the input saturation constraint problem in the design of the controller to reduce the system output tracking error.

In this paper, considering the model parameter uncertainty, the unknown bounded friction torque, the unknown bounded external disturbance and the control input saturation constraint problem, the dynamic equation of the variable stiffness joint is constructed. The coordinate transformation and feedback linearization method are used to transform the complex uncertain nonlinear system state space model into a pseudo linearized system model. The linear extended state observer is used to estimate the system state of the VSJ output link angular position and joint output stiffness. At the same time, a new adaptive controller based on neural state feedback method and robust control for adaptive input saturation compensation control are designed by RBFNN. This reduces the tracking error of the system and optimizes the VSJ output link angular position and joint output stiffness cooperative tracking control algorithm. Finally, the stability analysis based on Lyapunov function proves the final uniform bounded stability of the closed-loop system. The effectiveness of the proposed algorithm is verified by compare simulation and semi-physical experiments.

The paper is organized as follows: Section 2 presents the design of the space manipulator VSJ, its nonlinear dynamic model and its linear transformation with the feedback linearization result. The design of LESO, and the adaptive neural state feedback robust control algorithm with input saturation constraint based on RBFNN are depicted in Section 3, as well as the proof of stability and robustness. The simulation results are shown in Section 4, and the semi-physical experimental results are shown in Section 5. The discussions of the results of simulation and semi-physical experiments are shown in Section 6. Conclusions, discussions, and future works are provided in Section 7.

## 2. Structural Design and Dynamic Model Construction of Space Manipulator VSJ

### 2.1. Space Manipulator VSJ Design

The simplified model of the space manipulator is shown in Figure 1. The joints of the model have three degrees of freedom, called pitch, yaw, and roll, respectively. Together with the six degrees of freedom possessed by the base, the space robots with three joints in Figure 1 have a total 15 degrees of freedom. In order to reduce the model dimension and focus on the VSJ, only the single joint single degree of freedom model is dynamically modeled, when constructing the dynamic model by the Lagrange equation. In this paper, the VSJ model is constructed with reference to the VSJ model with link angle control motor and stiffness control motor mounted in series [16]. The prototype model of this model is shown in Figure 2.

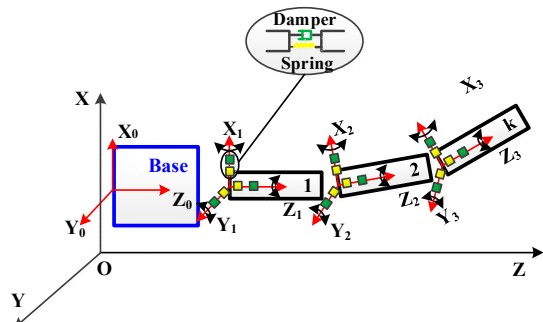

**Figure 1.** The simplified model of space manipulator.

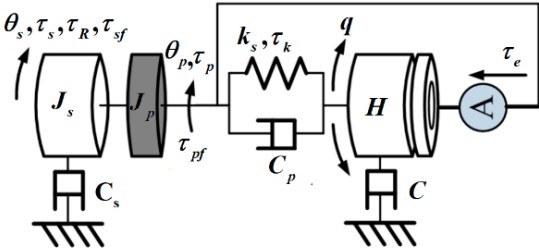

**Figure 2.** The prototype of variable stiffness joint (VSJ) model.

### 2.2. Dynamic Model Construction

Consider the parameter uncertainty, the unknown friction torque, the unknown external disturbance, and the input saturation constraint in the VSJ dynamics model. The actual system dynamics model is shown in (1).

$$\begin{cases} H_t\ddot{q} + C_t\dot{q} + \tau_k = \tau_e \\ J_{pt}\ddot{\theta}_p + C_{pt}\dot{\theta}_p - \tau_k + \tau_{pf} = \tau_p \quad ; \tau_R = \frac{2k_s n^2 \theta_s \phi^2 \Omega^3}{(\Omega - n\theta_s)^3}; \tau_k = K\phi = \frac{2k_s \delta_1^2 \Omega^2 \phi}{(\Omega - \delta_1)^2}; K = \frac{2k_s \delta_1^2 \Omega^2}{(\Omega - \delta_1)^2}; \delta_1 = n\theta_s; \\ J_{st}\ddot{\theta}_s + C_{st}\dot{\theta}_s + \tau_{sf} + \tau_R = \tau_s \end{cases} \tag{1}$$

The first row of the equations represents the dynamic equation of the space manipulator, the second row represents the dynamic equation of the joint angle control motor of VSJ, and the third row represents the dynamic equation of the joint stiffness control motor of VSJ. Where $H, C, J_p, C_p, J_s, C_s$ represent the system equivalent moment of inertia and equivalent friction damping coefficient of the joint of the space manipulator, the system equivalent moment of inertia and the equivalent friction coefficient of the joint angle control motor and the system equivalent moment of inertia and equivalent friction of the joint stiffness control motor Damping coefficient. $H_t, C_t, J_{pt}, C_{pt}, J_{st}, C_{st}$ represent the actual moment of inertia and the actual frictional damping coefficient of the joint of the space manipulator joint, the actual moment of inertia and the actual frictional damping coefficient of the joint angle control motor, and the

actual moment of inertia and the actual frictional damping coefficient of the joint stiffness control motor. We have $\Delta H = H_t - H$, $\Delta C = C_t - C$, $\Delta J_p = J_{pt} - J_p$, $\Delta C_p = C_{pt} - C_p$, $\Delta J_s = J_{st} - J_s$, $\Delta C_s = C_{st} - C_s$. $q$ represents the joint output angle. $\theta_p$ represents the joint main motor rotation angle. $\tau_k$ is the elastic moment, and $\tau_R$ is a reaction torque acting on the stiffness control unit due to elastic transfer. $\tau_p$ is the input torque provided by the joint angle control drive unit. $\tau_s$ is the input torque provided by the joint stiffness control drive unit. $\tau_{pf}$ and $\tau_{sf}$ is the parameter uncertainty, unknown bounded friction torque. $\tau_e$ is an unknown bounded external disturbance. $k_s$ is the stiffness coefficient of the stiffness control motor. $K$ is the output stiffness coefficient matrix of the joints $\Omega$ is a fixed value for the length of the chute in the internal lever of the joint. $\delta_1 = n\theta_s$ is the distance from the pivot position to the center of rotation of the joint. $\phi = q - \theta_p$ is the amount of deformation for the elastic transmission, whose range is [−0.35 rad, 0.35 rad ] according to the reference [15]. $\theta_s$ is the rotation angle of the stiffness control motor, and n is the gear ratio of the rack and pinion.

It can be seen from (1) that when the stiffness of the antagonist spring is selected, the relationship between the joint stiffness and the pivot position can be obtained. It can be seen that the joint stiffness has a large variation range when the pivot range is small. The joint stiffness can be continuously adjusted to meet different requirements in actual work. The system state-space model of the space manipulator VSJ is as follows.

$$\begin{cases} \dot{x} = f(x) + g(x)u + d_w(x) \\ y = h(x) \end{cases} ; u = \begin{bmatrix} u_p \\ u_s \end{bmatrix}; y = \begin{bmatrix} h_1(x) \\ h_2(x) \end{bmatrix} = \begin{bmatrix} q \\ k \end{bmatrix} = \begin{bmatrix} x_1 \\ \dfrac{2k_s\delta_1^2\Omega^2}{(\Omega-\delta_1)^2} \end{bmatrix} \tag{2}$$

$$x = [x_1, x_2, x_3, x_4, x_5, x_6]^T = \left[q, \dot{q}, \theta_p, \dot{\theta}_p, \theta_s, \dot{\theta}_s\right]^T \tag{3}$$

$$f(x) = \left[x_2, -\frac{C}{H}x_2 - \frac{\tau_k}{H}, x_4, -\frac{C_p}{J_p}x_4 + \frac{\tau_k}{J_p}, x_6, -\frac{C_s}{J_s}x_6 - \frac{\tau_R}{J_s}\right]^T ;$$

$$g(x) = \begin{bmatrix} g_p(x) & g_s(x) \end{bmatrix} = \begin{bmatrix} 0 & 0 & 0 & 1/J_p & 0 & 0 \\ 0 & 0 & 0 & 0 & 0 & 1/J_s \end{bmatrix}^T \tag{4}$$

Considering the existence of input saturation constraints, the system's control input variables need to be improved as follows.

$$sat(u) = \begin{bmatrix} sat(u_p) \\ sat(u_s) \end{bmatrix} = \begin{bmatrix} u_p + \Delta u_p \\ u_s + \Delta u_s \end{bmatrix}; sat(u_p) = \begin{cases} u_{pmax}; if & u_p \geq u_{pmax} \\ u_p; if & u_{pmin} < u_p < u_{pmax} \\ u_{pmin}; if & u_p \leq u_{pmin} \end{cases} ; sat(u_s) = \begin{cases} u_{smax}; if & u_s \geq u_{smax} \\ u_s; if & u_{smin} < u_s < u_{smax} \\ u_{smin}; if & u_s \leq u_{smin} \end{cases} \tag{5}$$

The composite disturbance in the VSJ's state space model is:

$$\begin{aligned} d_w(x) &= [d_1, d_2, d_3, d_4, d_5, d_6]^T \\ &= \left[0, -\frac{\Delta H}{H}\dot{x}_2 - \frac{\Delta C_p}{H}x_2 + \frac{\tau_e}{H}, 0, -\frac{\Delta J_p}{J_p}\dot{x}_4 - \frac{\Delta C_p}{J_p}x_4 + \frac{\tau_{pf}}{J_p}, 0, -\frac{\Delta J_s}{J_s}\dot{x}_6 - \frac{\Delta C_s}{J_s}x_6 - \frac{\tau_{sf}}{J_s}\right]^T \end{aligned} \tag{6}$$

**Assumption 1.** *Model parameter uncertainty in VSJ's system dynamics model, unknown friction torque and unknown external disturbances are bounded. Since the composite disturbance $d(x) \in R^6$ is a composite function of system state, unknown parameter perturbation, unknown friction torque, and unknown external disturbance, the composite disturbance $d(x) \in R^6$ is also bounded.*

The nonlinear system model constructed by (2) can be transformed into a linearized system model by coordinate transformation. Make $z = \left[z_q, z_k\right]^T$, $z_q = \left[z_{q1}, z_{q2}, z_{q3}, z_{q4}\right]$, $z_k = [z_{k1}, z_{k2}]$.

The nonlinear coordinate transformation of the high-order derivative containing the disturbance is performed in (7), so that the nonlinear system state-space model of the VSJ including the composite disturbance and the input saturation constraint is transformed into an integral linear chain of pseudo-linear systems with the matched aggregate disturbance and input saturation constraint.

$L_Y^n X$ represents the $n$ th Li derivative of the function $X$ with respect to the function $Y$, where $X$ can be $h_i(x), i = 1, 2$ and the Li derivative of them while, $Y$ can be $f(x)$, $g_p(x)$, $g_s(x)$, and $d(x)$.

$$
z_q = \begin{bmatrix} z_{q1} \\ z_{q2} \\ z_{q3} \\ z_{q4} \end{bmatrix} = \begin{bmatrix} h_1(x) \\ L_f h_1(x) + L_d h_1(x) \\ L_f^2 h_1(x) + L_d L_f h_1(x) + \frac{d}{dt}[L_d h_1(x)] \\ L_f^3 h_1(x) + L_d L_f^2 h_1(x) + \frac{d}{dt}[L_d L_f h_1(x)] + \frac{d^2}{dt^2}[L_d h_1(x)] \end{bmatrix}
$$

$$
z_k = \begin{bmatrix} z_{k1} \\ z_{k2} \end{bmatrix} = \begin{bmatrix} h_2(x) \\ L_f h_2(x) + L_d h_2(x) \end{bmatrix}; z = \begin{bmatrix} z_q \\ z_k \end{bmatrix}; \begin{cases} \dot{z} = Az + Bsat(v) + Bv_d \\ y_o = Cz \end{cases}; v_d = \begin{bmatrix} v_{dq} \\ v_{dk} \end{bmatrix} \tag{7}
$$

where

$$
v_{dq} = L_d L_f^3 h_1(x) + \frac{d}{dt}\left[L_d L_f^2 h_1(x)\right] + \frac{d^2}{dt^2}\left[L_d L_f h_1(x)\right] + \frac{d^3}{dt^3}[L_d h_1(x)]
$$

$$
v_{dk} = L_d L_f h_2(x) + \frac{d}{dt}[L_d h_1(x)]
$$

where the state matrix A, the input matrix B, and the output matrix C are as follows:

$$
A = \begin{bmatrix} 0 & 1 & 0 & 0 & 0 & 0 \\ 0 & 0 & 1 & 0 & 0 & 0 \\ 0 & 0 & 0 & 1 & 0 & 0 \\ 0 & 0 & 0 & 0 & 0 & 0 \\ 0 & 0 & 0 & 0 & 0 & 1 \\ 0 & 0 & 0 & 0 & 0 & 0 \end{bmatrix}; B = \begin{bmatrix} 0 & 0 \\ 0 & 0 \\ 0 & 0 \\ 1 & 0 \\ 0 & 0 \\ 0 & 1 \end{bmatrix}; C = \begin{bmatrix} 1 & 0 & 0 & 0 & 0 & 0 \\ 0 & 0 & 0 & 0 & 1 & 0 \end{bmatrix} \tag{8}
$$

$$
rank\left[C, CA, \ldots, CA^5\right]^T = rank\left[B, AB, \ldots, A^5 B\right] = 6 \tag{9}
$$

The system $\{A, B\}$ is controllable and the system $\{A, C\}$ is observable.

Combined with the feedback linearization method, in the case of input saturation constraints, the system's control input variables is shown in (10).

$$
v = \begin{bmatrix} v_q \\ v_k \end{bmatrix}; \Delta v = \begin{bmatrix} \Delta v_q \\ \Delta v_k \end{bmatrix}; sat(v) = v + \Delta v
$$

$$
\begin{cases} v_q = L_f^4 h_1(x) + L_{gp} L_f^3 h_1(x) u_p + L_{gs} L_f^3 h_1(x) u_s \\ v_k = L_f^2 h_2(x) + L_{gp} L_f h_2(x) u_p + L_{gs} L_f h_2(x) u_s \\ \Delta v_q = L_{gp} L_f^3 h_1(x) \Delta u_p + L_{gs} L_f^3 h_1(x) \Delta u_s \\ \Delta v_k = L_{gp} L_f h_2(x) \Delta u_p + L_{gs} L_f h_2(x) \Delta u_s \end{cases} \tag{10}
$$

$$
sat(v) = \begin{bmatrix} L_{gk} L_f^3 h_1(x) & L_{gs} L_f^3 h_1(x) \\ L_{gk} L_f h_2(x) & L_{gs} L_f h_2(x) \end{bmatrix} \begin{bmatrix} sat(u_q) \\ sat(u_s) \end{bmatrix} + \begin{bmatrix} L_f^4 h_1(x) \\ L_f^2 h_2(x) \end{bmatrix}
$$

The determinant of the feedback linearization decoupling matrix $G(x)$ of the VSJ constructed in (10) is as shown in equation (11).

$$
G(x) = \begin{bmatrix} L_{gp} L_f^3 h_1(x) & L_{gs} L_f^3 h_1(x) \\ L_{gp} L_f h_2(x) & L_{gs} L_f h_2(x) \end{bmatrix}; det(G(x)) = \frac{8 k_s^2 n^4 x_5^3 \Omega^5}{J_p J_s H (\Omega - n x_5)^2} \tag{11}
$$

The joint output stiffness defined in this paper can be adjusted as $0 < K < +\infty$. According to the description of (5), it can be seen that the determinant of the decoupling matrix $G(x)$ will always be non-zero, that is, the decoupling matrix is always non-singular.

## 3. Control Algorithm Construction

### 3.1. The Linear Extended State Observer Design

The Linear Extended State Observer is used to estimate unknown system states, that is $z_{q2}, z_{q3}, z_{q4}$, and $z_{k2}$, and also the matched aggregate disturbance $v_{dq}$ and $v_{dk}$.

**Assumption 2.** *The expanded state variables set as $z_{q5} = v_{dq}, z_{k3} = v_{dk}$ are bounded and differentiable, which are the functions of system state variables and unknown variables in the system. That is, there are $V_{dq} > 0$ and $V_{dk} > 0$, so that $|v_{dq}| \leq V_{dq}$ and $|v_{dk}| \leq V_{dk}$ are established. The variables $dz_{q5}$ and $dz_{k3}$ represent the first derivative of $z_{q5}$ and $z_{k5}$ with respect to time t, respectively. They are also bounded.*

The resulting extended pseudo linear system is shown as follows.

$$\begin{cases} \dot{z}_e = A_e z_e + B_e sat(v) + B_{de} d_e \\ \qquad\qquad y_e = C_e z_e \end{cases} ;$$ (12)

The related matrix description of (12) is as follows.

$$d_e = \begin{bmatrix} dz_{q5} \\ dz_{k3} \end{bmatrix}; C_e = \begin{bmatrix} 1 & 0 & 0 & 0 & 0 & 0 & 0 & 0 \\ 0 & 0 & 0 & 0 & 1 & 0 & 0 & 0 \end{bmatrix}$$ (13)

$$z_e = \begin{bmatrix} z_{q1} \\ z_{q2} \\ z_{q3} \\ z_{q4} \\ z_{k1} \\ z_{k2} \\ z_{q5} \\ z_{k3} \end{bmatrix}; A_e = \begin{bmatrix} 0 & 1 & 0 & 0 & 0 & 0 & 0 & 0 \\ 0 & 0 & 1 & 0 & 0 & 0 & 0 & 0 \\ 0 & 0 & 0 & 1 & 0 & 0 & 0 & 0 \\ 0 & 0 & 0 & 0 & 0 & 0 & 0 & 0 \\ 0 & 0 & 0 & 0 & 0 & 1 & 0 & 0 \\ 0 & 0 & 0 & 0 & 0 & 0 & 0 & 1 \\ 0 & 0 & 0 & 0 & 0 & 0 & 0 & 0 \\ 0 & 0 & 0 & 0 & 0 & 0 & 0 & 0 \end{bmatrix}; B_e = \begin{bmatrix} 0 & 0 \\ 0 & 0 \\ 0 & 0 \\ 1 & 0 \\ 0 & 0 \\ 0 & 1 \\ 0 & 0 \\ 0 & 0 \end{bmatrix}; B_{de} = \begin{bmatrix} 0 & 0 \\ 0 & 0 \\ 0 & 0 \\ 0 & 0 \\ 0 & 0 \\ 0 & 0 \\ 1 & 0 \\ 0 & 1 \end{bmatrix}$$ (14)

The designed linear extended state observer (LESO) is shown below.

$$\begin{cases} \dot{\hat{z}}_e = A_e \hat{z}_e + B_e sat(v) + L_e C_e (z_e - \hat{z}_e) \\ \qquad\qquad \hat{y}_e = C_e \hat{z}_e \\ \hat{z}_{q1}(0) = z_{q1}(0); \hat{z}_{k1}(0) = z_{k1}(0) \end{cases} ; \quad \begin{aligned} \hat{z}_e &= \left[ \hat{z}_{q1}, \hat{z}_{q2}, \hat{z}_{q3}, \hat{z}_{q4}, \hat{z}_{k1}, \hat{z}_{k2}, \hat{z}_{q5}, \hat{z}_{k3} \right]^T \\ L_e &= \begin{bmatrix} l_1 & l_2 & l_3 & l_4 & 0 & 0 & l_5 & 0 \\ 0 & 0 & 0 & 0 & l_6 & l_7 & 0 & l_8 \end{bmatrix}^T \end{aligned}$$ (15)

where $\hat{z}_e$ represents the estimate of $z_e$, and $L_e \in R^{8 \times 2}$ is the observed gain matrix of LESO. The characteristic equation of LESO is described as follows:

$$\begin{cases} \xi_q^5 + l_1 \xi_q^4 + l_2 \xi_q^3 + l_3 \xi_q^2 + l_4 \xi_q + l_5 = 0 \\ \xi_k^3 + l_6 \xi_k^2 + l_7 \xi_k + l_8 = 0 \end{cases}$$ (16)

The choice of LESO's observation gain can be determined by the pole placement theory. In order to obtain a good dynamic process and noise suppression capability, the characteristic equation is set as shown in the following equation, where $\lambda_q$ and $\lambda_k$ are both negative values. Their values can be selected using the bandwidth concept [38].

$$(\xi_q - \lambda_q)^5 = 0; (\xi_k - \lambda_k)^3 = 0$$ (17)

The state estimation error of LESO is defined as:

$$e_{lo} = \begin{bmatrix} e_{qk} \\ e_{qke} \end{bmatrix}; e_{qk} = \begin{bmatrix} e_{q1} \\ e_{q2} \\ e_{q3} \\ e_{q4} \\ e_{k1} \\ e_{k2} \end{bmatrix} = \begin{bmatrix} z_{q1} - \hat{z}_{q1} \\ z_{q2} - \hat{z}_{q2} \\ z_{q3} - \hat{z}_{q3} \\ z_{q4} - \hat{z}_{q4} \\ z_{k1} - \hat{z}_{k1} \\ z_{k2} - \hat{z}_{k2} \end{bmatrix}; e_{qke} = \begin{bmatrix} e_{q5} \\ e_{k3} \end{bmatrix} = \begin{bmatrix} z_{q5} - \hat{z}_{q5} \\ z_{k3} - \hat{z}_{k3} \end{bmatrix}$$ (18)

The LESO state estimation error system is as follows:

$$\dot{e}_{lo} = (A_e - L_e C_e)e_{lo} + B_{le}u_{le} = (A_e - L_e C_e)e_{lo} + d_{le} \tag{19}$$

where $B_{le} \in R^{8\times4}, u_{le} \in R^{4\times1}, d_{le} \in R^{8\times1}$ are shown in (20).

$$B_{le} = \begin{bmatrix} 0 & 0 & 0 & 1 & 0 & 0 & 0 & 0 \\ 0 & 0 & 0 & 0 & 0 & 1 & 0 & 0 \\ 0 & 0 & 0 & 0 & 0 & 0 & 1 & 0 \\ 0 & 0 & 0 & 0 & 0 & 0 & 0 & 1 \end{bmatrix}^T ; u_{le} = \begin{bmatrix} 0 \\ 0 \\ dz_{q5} \\ dz_{k3} \end{bmatrix} ; d_{le} = \begin{bmatrix} 0,0,0,0,0,0,dz_{q5},dz_{k3} \end{bmatrix}^T \tag{20}$$

Since System $\{A, C\}$ is observable, System $\{A_e, C_e\}$ is also observable. $d_{le}$ is bounded.

**Assumption 3.** *If $A_e - L_e C_e$ is Hurwitz, then (19) will have BIBO stability. That is, there are $\varepsilon_{dp} \geq 0$ and $\varepsilon_{dk} \geq 0$, so that $\widetilde{z}_{q5} = |z_{q5} - \hat{z}_{q5}| \leq \varepsilon_{dp}$ and $\widetilde{z}_{k3} = |z_{k3} - \hat{z}_{k3}| \leq \varepsilon_{dk}$ are established.*

*3.2. Neural State Feedback Adaptive Robust Control Based on RBFNN*

3.2.1. Design of Neural State Feedback Adaptive Robust Control Based on RBFNN

**Assumption 4.** *The reference joint rotation angle $q_d$ and stiffness $k_d$ and their derivative are smooth bounded.*

Define the tracking error vector as:

$$E = [E_q, E_k] = \left[ e_q, \dot{e}_q, \ddot{e}_q, \dddot{e}_q, e_k, \dot{e}_k \right]^T \tag{21}$$

where $e_q$ and $e_k$ are the output errors of joint angle and joint stiffness as follows:

$$\begin{cases} e_q = z_{q1} - q_d \\ e_k = z_{k1} - k_d \end{cases} \tag{22}$$

According to (7), when the disturbance term $v_d$ is not considered, the tracking error system can be expressed as follows:

$$\begin{aligned} \dot{E} &= AE^T + B\left(v_f - z_d^{(n)}\right); \\ e &= \begin{bmatrix} e_q \\ e_k \end{bmatrix} = C \begin{bmatrix} E_q \\ E_k \end{bmatrix} \end{aligned} \tag{23}$$

where $v_f = \left[ v_{qf}, v_{kf} \right]^T, z_d^{(n)} = \left[ q_d^{(4)}, k_d^{(2)} \right]^T$. The feedback controller $v_f$ can be expressed as:

$$v_f = z_d^{(n)} - \Lambda E \tag{24}$$

where $\Lambda = \begin{bmatrix} \Lambda_q & 0 \\ 0 & \Lambda_k \end{bmatrix}, \Lambda_q = [\Lambda_{q1}, \Lambda_{q2}, \Lambda_{q3}, \Lambda_{q4}]^T, \Lambda_k = [\Lambda_{k1}, \Lambda_{k2}]^T$. The choice of $\Lambda_q$ and $\Lambda_k$ should make sure that the corresponding characteristic polynomial is Hurwitz. Substituting (24) into (23), the tracking error system can be expressed as follows:

$$\dot{E} = \begin{bmatrix} \dot{E}_q \\ \dot{E}_k \end{bmatrix} = \left( A - B\Lambda^T \right) \begin{bmatrix} E_q \\ E_k \end{bmatrix} \tag{25}$$

**Lemma 1.** *Assume a positive definite symmetry matrix $P_1$ is given. There must be a positive definite matrix $Q_1$ that satisfies (26) as follows:*

$$\left( A - B\Lambda^T \right)^T P_1 + P_1 \left( A - B\Lambda^T \right) = -Q_1 \tag{26}$$

*where* $A = \begin{bmatrix} A_{q1} & 0 \\ 0 & A_{k1} \end{bmatrix} \in R^{6\times6}$, $A_q = \begin{bmatrix} 0 & 1 & 0 & 0 \\ 0 & 0 & 1 & 0 \\ 0 & 0 & 0 & 1 \\ 0 & 0 & 0 & 0 \end{bmatrix}$, $A_k = \begin{bmatrix} 0 & 1 \\ 0 & 0 \end{bmatrix}$, $B = \begin{bmatrix} B_{q1} & 0 \\ 0 & B_{k1} \end{bmatrix} \in R^{2\times6}$,

$B_q = \begin{bmatrix} 0 \\ 0 \\ 0 \\ 1 \end{bmatrix}$, $B_k = \begin{bmatrix} 0 \\ 1 \end{bmatrix}$, $P_1 = \begin{bmatrix} P_{q1} & 0 \\ 0 & P_{k1} \end{bmatrix} \in R^{6\times6}$, $P_{q1} \in R^{4\times4}$, $P_{k1} \in R^{2\times2}$, $Q_1 = \begin{bmatrix} Q_{q1} & 0 \\ 0 & Q_{k1} \end{bmatrix} \in R^{6\times6}$,

$Q_{q1} \in R^{4\times4}$, $Q_{k1} \in R^{2\times2}$.

Define the Lyapunov function as:

$$V_1 = \tfrac{1}{2}E_q^T P_{q1} E_q; V_2 = \tfrac{1}{2}E_k^T P_{k1} E_k \tag{27}$$

The derivatives of $V_1$ and $V_2$ with respect to time t are as follows:

$$\dot{V}_1 = \tfrac{1}{2}\dot{E}_q^T P_{q1} E_q + \tfrac{1}{2}E_q^T P_{q1}\dot{E}_q; \dot{V}_2 = \tfrac{1}{2}\dot{E}_k^T P_{k1} E_k + \tfrac{1}{2}E_k^T P_{k1}\dot{E}_k \tag{28}$$

Substituting (25) and into (28), The derivatives of $V_1$ and $V_2$ can be changed into:

$$
\begin{aligned}
\dot{V}_1 &= \tfrac{1}{2}\dot{E}_q^T P_{q1} E_q + \tfrac{1}{2}E_q^T P_{q1}\dot{E}_q \\
&= \tfrac{1}{2}\left(\left(A_q - B_q\Lambda_q^T\right)E_q\right)^T P_{q1}E_q + \tfrac{1}{2}E_q^T P_{q1}\left(\left(A_q - B_q\Lambda_q^T\right)E_q\right) \\
&= \tfrac{1}{2}E_q^T\left[\left(A_q - B_q\Lambda_q^T\right)^T P_{q1} + P_{q1}\left(A_q - B_q\Lambda_q^T\right)\right]E_q \\
\dot{V}_2 &= \tfrac{1}{2}\dot{E}_k^T P_{k1} E_k + \tfrac{1}{2}E_k^T P_{k1}\dot{E}_k \\
&= \tfrac{1}{2}\left(\left(A_k - B_k\Lambda_k^T\right)E_k\right)^T P_{k1}E_k + \tfrac{1}{2}E_k^T P_{k1}\left(\left(A_k - B_k\Lambda_k^T\right)E_k\right) \\
&= \tfrac{1}{2}E_k^T\left[\left(A_k - B_k\Lambda_k^T\right)^T P_{k1} + P_{k1}\left(A_k - B_k\Lambda_k^T\right)\right]E_k
\end{aligned}
\tag{29}
$$

As can be seen from (26), the above formula can be expressed as:

$$\dot{V}_1 = -\tfrac{1}{2}\dot{E}_q^T Q_{q1} E_q \leq 0; \dot{V}_2 = -\tfrac{1}{2}\dot{E}_k^T Q_{k1} E_k \leq 0 \tag{30}$$

Thus, the above system is Lyapunov stable.

The control input can be rewritten when considering controller robustness and input saturation compensation:

$$v = v_f + v_{fns} + v_r + v_n - v_s \tag{31}$$

where $v_{fns}$ is the output of the state feedback controller. $v_r$ is a robust controller. $v_s$ is the input saturation compensator. $v_n$ is the compensation function for unknown external disturbances. According to (10), $v_q = v_{qf} + v_{qfns} + v_{qn} + v_{qr} - v_{qs}$ and $v_k = v_{kf} + v_{kfns} + v_{kn} + v_{kr} - v_{ks}$.

As can be seen from the LESO, the compensation function for unknown external disturbances can be written as:

$$v_n = \left[-\hat{v}_{qn}, -\hat{v}_{kn}\right]^T = -\hat{z}_n = \left[-\hat{z}_{q5}, -\hat{z}_{k3}\right]^T \tag{32}$$

The output of the state feedback controller $v_{fns}$ can be designed as follows:

$$v_{fns} = -Tz = -T\left[z_q, z_k\right]^T, T = \left[T_{q1}(z), T_{q2}(z), T_{q3}(z), T_{q4}(z), T_{k1}(z), T_{k2}(z)\right] \tag{33}$$

$T$ is a state feedback gain matrix.

The RBFNN has a simple structure, fast convergence, better generalization ability, and can approximate any continuous nonlinear function with higher precision. The structure adopts parallel processing mechanism and has strong fault tolerance. Based on the better approximation characteristics of RBFNN, the unknown function can be approximated with arbitrary precision. Therefore, RBFNN can better meet the requirements of system control with uncertainties.

Using the RBFNN approximates the state feedback gain matrix, as shown in (34):

$$\hat{T} = \hat{W}^T \phi(z) \tag{34}$$

where $\hat{W} = \left[\hat{W}_q, \hat{W}_k\right] \in R^{6 \times n}$ is the estimated value of the weight matrix of the RBFNN. $\phi(z) = \left[\phi(z_q), \phi(z_k)\right]^T$ is also chosen to be Gaussian RBFs just like above:

$$\phi_j(z_{qi}) = \exp\left(\frac{\|z_{qi} - c_{\phi qj}\|}{2b_{\phi qj}^2}\right), \phi_j(z_{ki}) = \exp\left(\frac{\|z_{ki} - c_{\phi kj}\|}{2b_{\phi kj}^2}\right);$$
$$qi = q1, q2, q3, q4, ki = k1, k2; j = 1, \cdots, n; \tag{35}$$

where $z_{qi}$ and $z_{ki}$ are just the same as above. $c_{\phi qj}$ and $c_{\phi kj}$ are the are centers of the Gaussian functions. $b_{\phi qj}$ and $b_{\phi kj}$ are the are widths of the Gaussian functions.

The ideal approximation of the state feedback gain matrix $T$ by the RBFNN can be expressed as follows:

$$T = W^T \phi(z) + \varepsilon_k \tag{36}$$

**Assumption 5.** *For any given sufficiently small positive number* $\varepsilon_{km} = \left[\varepsilon_{kmq}, \varepsilon_{kmk}\right]$, *the optimal weight matrix* $W^* = \left[W_q^*, W_k^*\right]$ *can always be found. the approximation error should satisfy:*

$$\sup_{z_q \in \Omega_{z_q}} \left|W_q^T \phi(z_q) - W_q^{*T} \phi(z_q)\right| = \|\varepsilon_{kq}\| \leq \varepsilon_{kmq}$$
$$\sup_{z_k \in \Omega_{z_k}} \left|W_k^T \phi(z_k) - W_k^{*T} \phi(z_k)\right| = \|\varepsilon_{kk}\| \leq \varepsilon_{kmk} \tag{37}$$

*where* $z_q \in \Omega_{z_q} = \left\{z_q \middle| \|z_q\| \leq M_q, M_q > 0\right\}, z_k \in \Omega_{z_k} = \{z_k \| \|z_k\| \leq M_k, M_k > 0\}$.

$\hat{W}^T \phi(z) - W^{*T} \phi(z) = \varepsilon_k, \varepsilon_k = \left[\varepsilon_{kq}, \varepsilon_{kk}\right]^T$ *is the neural network approximation error, and the optimal weight vector* $W^*$ *is defined as follows:*

$$W^* = \arg \min_{W \in \Omega_W} \left\{\sup_{z \in \Omega_z} \left|W^T \phi(z) - \hat{W}^T \phi(z)\right|\right\} = \arg \min_{W \in \Omega_W} \left\{\sup_{z \in \Omega_z} \left|\widetilde{W}^T \phi(z)\right|\right\} \tag{38}$$

According to Assumption 5, (39) can be obtained as follows:

$$\widetilde{W}^T \phi(z) = W^T \phi(z) - \hat{W}^T \phi(z) \tag{39}$$

Using the RBFNN to approximate the state feedback gain matrix, the output of the state feedback controller can be obtained as:

$$v_{fns} = -Tz = -\left[\hat{W}_q^T \phi_q, \hat{W}_k^T \phi_k\right]\begin{bmatrix} z_q \\ z_k \end{bmatrix} \tag{40}$$

Combined with (7), (31), and (33), (41) can be obtained:

$$\dot{z} = (A - BT)z + Bv_c \tag{41}$$

where $v_c = v_f + v_r + v_n - v_s$. The characteristic equation of $A - BT$ is as follows:

$$|A - BT - \lambda I| = 0 \tag{42}$$

where $\lambda = [\lambda_1, \lambda_2, \ldots, \lambda_6]$ is a feature vector. According to the design principle of the state feedback gain matrix, $T$ should satisfy with (43)

$$|A - BT - \lambda I| = |A - \lambda I| \tag{43}$$

(41) can be rewritten as

$$\dot{z} = Az + Bv_c + Bv_d = Az + Bv \tag{44}$$

According to, (44) can be expressed as:

$$\dot{z} = Az + B\widetilde{T}z + Bv_c^* \tag{45}$$

where $\widetilde{v}_d = v_n + v_d$, $v_c^* = \widetilde{v}_d + v_f + v_r$.

And then, The tracking error system (25) can be rewritten as follows:

$$\dot{E} = AE + B\left(\widetilde{v}_d - \Lambda^T E + v_r\right) \tag{46}$$

Considering (15), there is:

$$\dot{\hat{z}} = A\hat{z} + Bv_c^* + L_l C(z - \hat{z}) \tag{47}$$

where $L_l = \begin{bmatrix} L_{lq} \\ L_{lk} \end{bmatrix} = \begin{bmatrix} l_1 & l_2 & l_3 & l_4 & 0 & 0 \\ 0 & 0 & 0 & 0 & l_6 & l_7 \end{bmatrix}^T$. According to (45) and (47), the state error can be defined as:

$$E_m = \left[E_{qm}, E_{km}\right]^T = z - \hat{z} = \left[z_q - \hat{z}_q, z_k - \hat{z}_k\right]^T = \left[\widetilde{z}_{q1}, l_1\widetilde{z}_{q1}, l_2\widetilde{z}_{q1}, l_3\widetilde{z}_{q1}, \widetilde{z}_{k1}, l_6\widetilde{z}_{k1}\right]^T \tag{48}$$

It can be seen that the state error equation can be expressed as follows:

$$\dot{E}_m = \dot{z} - \dot{\hat{z}} = (A - L_l C)E_m + B\widetilde{T}z = (A - L_l C)\hat{z} + B\left(\widetilde{W}^T \phi(z)\right)z \tag{49}$$

where $C = \begin{bmatrix} C_q & 0 \\ 0 & C_k \end{bmatrix} \in R^{2\times6}$, $C_q = [1, 0, 0, 0]$, $C_k = [1, 0]$.

**Lemma 2.** *Assuming that a positive definite symmetric matrix $P_2$ is given, there must be a positive definite matrix $Q_2$ that satisfies:*

$$A^T P_2 + P_2 A = -Q_2 \tag{50}$$

*where* $P_2 = \begin{bmatrix} P_{q2} & 0 \\ 0 & P_{k2} \end{bmatrix} \in R^{6\times6}$, $Q_2 = \begin{bmatrix} Q_{q2} & 0 \\ 0 & Q_{k2} \end{bmatrix} \in R^{6\times6}$.

The adaptive laws of $\hat{W}$ is:

$$\dot{\hat{W}} = \gamma_w \phi(z)^T B^T P_2 E_m z \tag{51}$$

where $\dot{\hat{W}} = \left[\dot{\hat{W}}_q, \dot{\hat{W}}_k\right]$, $\gamma_w = \left[\gamma_{qw}, \gamma_{kw}\right]$ is a positive constants matrix.

The output of the robust controller is:

$$v_r = v_{vs} + v_{rc} \tag{52}$$

$$v_{vs} = -\kappa_s \text{sgn}\left(B^T P_1 E\right) \tag{53}$$

$$v_{rc} = -\eta B^T P_1 E \tag{54}$$

where $v_{vs} = \left[v_{vsp}, v_{vsk}\right]^T$, $\kappa_s = \left[\kappa_{qs}, \kappa_{ks}\right]^T$; $\kappa_{qs} \geq \|\varepsilon_{dp} + \varepsilon_{qs}\|$; $\kappa_{ks} \geq \|\varepsilon_{dk} + \varepsilon_{ks}\|$, $v_{rc} = \left[v_{rcp}, v_{rck}\right]^T$, $\eta = \left[\eta_q, \eta_k\right]^T$; $\eta_q > 0, \eta_k > 0$. $v_{vs}$ is used to eliminate the influence of neural network approximation errors and the effects of external disturbances. $v_{rc}$ is used to improve the robust performance of the

system. The saturation compensation $v_s$, which is also an estimate of the saturation function, can be written as:

$$\hat{v}_s = \hat{U}^T \varphi(\xi) \tag{55}$$

The saturation function can be expressed as

$$v_s = U^T \varphi(\xi) + \varepsilon_s \tag{56}$$

$U^T \varphi(\xi) = \left[ U_p^T \varphi(\xi_p), U_k^T \varphi(\xi_k) \right] \in R^{n \times n}$ is the unknown ideal weight vector.
$v_s = \left[ v_s, v_p \right]^T$, $\xi = \begin{bmatrix} \xi_q \\ \xi_k \end{bmatrix} = \begin{bmatrix} q_d & E_q \\ k_d & E_k \end{bmatrix}$. $\varphi(\xi) = \left[ \varphi_1(\xi), \varphi_2(\xi), \dots, \varphi_n(\xi) \right]^T \in R^n$ is the basis function vector. Select the Gaussian function:

$$\varphi_j(\xi_{qi}) = exp\left( \frac{\|\xi_{qi} - c_{\varphi qj}\|^2}{2b^2_{\varphi qj}} \right), \varphi_j(\xi_{ki}) = exp\left( \frac{\|\xi_{ki} - c_{\varphi kj}\|^2}{2b^2_{\varphi kj}} \right);$$
$$qi = q1, q2, q3, q4, q5, ki = k1, k2, k3; j = 1, 2, \cdots, n \tag{57}$$

where $c_{\varphi qj}$ and $c_{\varphi kj}$ are the are centers of the Gaussian functions. $b_{\varphi qj}$ and $b_{\varphi kj}$ are the are widths of the Gaussian functions. $\varepsilon_s$ is the reconstruction error. If n is large enough, RBFNN can approximate any continuous nonlinear function with arbitrary precision. According to Assumption 5, there is $\widetilde{U} = U - \hat{U}$. Define the approximation error as:

$$\widetilde{v}_s = \widetilde{U}^T \varphi(\xi) + \varepsilon_s = U^T \varphi(\xi) - \hat{U}^T \varphi(\xi) + \varepsilon_s \tag{58}$$

where $\varepsilon_s = \left[ \varepsilon_{ps}, \varepsilon_{ks} \right]$. The adaptive laws of $\hat{U}$ is:

$$\dot{\hat{U}} = \gamma_q \varphi(\xi)^T B^T P_1 E \tag{59}$$

where $\dot{\hat{U}} = \left[ \dot{\hat{U}}_q, \dot{\hat{U}}_k \right]$, $\gamma_q = \left[ \gamma_{qu}, \gamma_{qu} \right]$ is a positive constants matrix. (60) can be rewritten as:

$$\dot{z} = \left( A - B\hat{T} \right) z + B\left( \widetilde{v}_d - \Lambda^T E + v_r + \widetilde{U}^T \varphi(\xi) + \varepsilon_s \right) = \left( A - B\hat{T} \right) z + Bv_c^* \tag{60}$$

In summary, the control design is as follows:

$$v = \begin{bmatrix} v_q \\ v_k \end{bmatrix} = \begin{bmatrix} -\hat{W}_q^T \phi_q(z) z_q - \hat{z}_{q5} + q_d^{(4)} - \Lambda_q^T E_q - \eta_q B_q^T P_{q1} E_q - \kappa_{qs} sgn\left( B_q^T P_{q1} E_q \right) - \hat{U}_q^T \varphi_q(\xi_q) \\ -\hat{W}_k^T \phi_k(z) z_k - \hat{z}_{k3} + k_d^{(2)} - \Lambda_k^T E_k - \eta_k B_k^T P_{k1} E_k - \kappa_{ks} sgn\left( B_k^T P_{k1} E_k \right) - \hat{U}_k^T \varphi_k(\xi_k) \end{bmatrix} \tag{61}$$

(62) can be obtained by the feedback linearization method.

$$\begin{bmatrix} u_p \\ u_s \end{bmatrix} = \begin{bmatrix} L_{gp} L_f^3 h_1(x) & L_{gs} L_f^3 h_1(x) \\ L_{gp} L_f h_2(x) & L_{gs} L_f h_2(x) \end{bmatrix}^{-1} \begin{bmatrix} v_q \\ v_k \end{bmatrix} - \begin{bmatrix} L_f^4 h_1(x) \\ L_f^2 h_2(x) \end{bmatrix} \tag{62}$$

Adding the anti-saturation measures, the control law can be:

$$\begin{bmatrix} sat(u_p) \\ sat(u_s) \end{bmatrix} = \begin{bmatrix} L_{gp} L_f^3 h_1(x) & L_{gs} L_f^3 h_1(x) \\ L_{gp} L_f h_2(x) & L_{gs} L_f h_2(x) \end{bmatrix}^{-1} \begin{bmatrix} sat(v_q) \\ sat(v_k) \end{bmatrix} - \begin{bmatrix} L_f^4 h_1(x) \\ L_f^2 h_2(x) \end{bmatrix} \tag{63}$$

### 3.2.2. Stability Analysis of State Feedback Adaptive Robust Controller Based on RBFNN

Defining the Lyapunov function as:

$$V_3 = \frac{1}{2} E_q^T P_{q1} E_q + \frac{1}{2} E_{qm}^T P_{q2} E_{qm} + \frac{1}{2\gamma_{qw}} \widetilde{W}_q^T \widetilde{W}_q + \frac{1}{2\gamma_{qu}} \widetilde{U}_q^T \widetilde{U}_q \tag{64}$$

The derivatives of $V_3$ with respect to time t is as follows:

$$\dot{V}_3 = \tfrac{1}{2}\dot{E}_q{}^T P_{q1} E_q + \tfrac{1}{2} E_q{}^T P_{q1}\dot{E}_q + \tfrac{1}{2}\dot{E}_{qm}{}^T P_{q2} E_{qm} + \tfrac{1}{2} E_{qm}{}^T P_{q2}\dot{E}_{qm} + \tfrac{1}{\gamma_{qw}}\widetilde{W}_q{}^T\dot{\widetilde{W}}_q + \tfrac{1}{\gamma_{qu}}\widetilde{U}_q{}^T\dot{\widetilde{U}}_q$$

$$\begin{aligned}
&= \tfrac{1}{2}\Big(A_q E_q + B_q\big(\widetilde{v}_{dp} - \Lambda^T E + \widetilde{U}_q{}^T \varphi_p(\xi) + \varepsilon_{qs} + v_{qr}\big)\Big)^T P_{q1} E_q \\
&\quad + \tfrac{1}{2} E_q{}^T P_{q1}\Big(A_q E_q + B_q\big(\widetilde{v}_{dp} - \Lambda^T E + \widetilde{U}_q{}^T \varphi_p(\xi) + \varepsilon_{qs} + v_{qr}\big)\Big) \\
&\quad + \tfrac{1}{2}\Big(A_q E_{qm} + B_q\big(\widetilde{W}_q{}^T \phi_q(z)\big)z_q\Big)^T P_{q2} E_{qm} \\
&\quad + \tfrac{1}{2} E_{qm}{}^T P_{q2}\Big(A_q E_{qm} + B_q\big(\widetilde{W}_q{}^T \phi_q(z)\big)z_q\Big) + \tfrac{1}{\gamma_{qw}}\widetilde{W}_q{}^T\dot{\widetilde{W}}_q + \tfrac{1}{\gamma_{qu}}\widetilde{U}_q{}^T\dot{\widetilde{U}}_q
\end{aligned}$$

$$\begin{aligned}
&= \tfrac{1}{2} E_q{}^T\Big[\big(A_q - B_q\Lambda_q{}^T\big)^T P_{q1} + P_{q1}\big(A_q - B_q\Lambda_q{}^T\big)\Big]E_q + \tfrac{1}{2} E_{qm}{}^T\big(A_q{}^T P_{q2} + P_{q2} A_q\big)E_{qm} \\
&\quad + \phi_q(z)^T\widetilde{W}_q B_q{}^T P_{q2} E_{qm} z_q + \varphi_q(\xi)^T\widetilde{U}_q B_q{}^T P_{q1} E_q + \big(\widetilde{v}_{dp} + v_{vsq} + \varepsilon_{qs}\big)B_q{}^T P_{q1} E_q \\
&\quad + v_{rcq} B_q{}^T P_{q1} E_q + \tfrac{1}{\gamma_{qw}}\widetilde{W}_q{}^T\dot{\widetilde{W}}_q + \tfrac{1}{\gamma_{qu}}\widetilde{U}_q{}^T\dot{\widetilde{U}}_q
\end{aligned}$$

$$\begin{aligned}
&= -\tfrac{1}{2}\dot{E}_q{}^T Q_{q1} E_q - \tfrac{1}{2}\dot{E}_{qm}{}^T Q_{q2} E_{qm} + \varphi_q(\xi)^T\widetilde{U}_q B_q{}^T P_{q1} E_q \\
&\quad + \phi_q(z)^T\widetilde{W}_q B_q{}^T P_{q2} E_{qm} z_q + \big(\widetilde{v}_{dp} + v_{vsq} + \varepsilon_{qs}\big)B_q{}^T P_{q1} E_q \\
&\quad + v_{rcq} B_q{}^T P_{q1} E_q + \tfrac{1}{\gamma_{qw}}\widetilde{W}_q{}^T\dot{\widetilde{W}}_q + \tfrac{1}{\gamma_{qu}}\widetilde{U}_q{}^T\dot{\widetilde{U}}_q
\end{aligned} \tag{65}$$

Since $\dot{\widetilde{W}}_q = -\dot{\hat{W}}_q$ and $\dot{\widetilde{U}}_q = -\dot{\hat{U}}_q$, combined with the adaptive law, and substituting (53) and (54) into the above formula, the following conclusions can be drawn:

$$\begin{aligned}
\dot{V}_3 &= -\tfrac{1}{2}\dot{E}_q{}^T Q_{q1} E_q - \tfrac{1}{2}\dot{E}_{qm}{}^T Q_{q2} E_{qm} + \big(v_{vsq} + \widetilde{v}_{dp} + \varepsilon_{qs}\big)B_q{}^T P_{q1} E_q + v_{rcq} B_q{}^T P_{q1} E_q \\
&\leq -\tfrac{1}{2}\dot{E}_q{}^T Q_{q1} E_q - \tfrac{1}{2}\dot{E}_{qm}{}^T Q_{q2} E_{qm} \\
&\quad + \big(\varepsilon_{dp} + \varepsilon_{qs} - \kappa_{qvd}\big)sgn\big(B_q{}^T P_{q1} E_q\big)B_q{}^T P_{q1} E_q - \eta_q B_q{}^T P_{q1} E_q\big(B_q{}^T P_{q1} E_q\big) \\
&\leq -\tfrac{1}{2}\dot{E}_q{}^T Q_{q1} E_q - \tfrac{1}{2}\dot{E}_{qm}{}^T Q_{q2} E_{qm} \leq 0
\end{aligned} \tag{66}$$

So $V_3 > 0$, $\dot{V}_3 \leq 0$. Defining the Lyapunov function as:

$$V_4 = \frac{1}{2} E_k{}^T P_{k1} E_k + \frac{1}{2} E_{km}{}^T P_{k2} E_{km} + \frac{1}{2\gamma_{kw}}\widetilde{W}_k{}^T\widetilde{W}_k + \frac{1}{2\gamma_{ku}}\widetilde{U}_k{}^T\widetilde{U}_k \tag{67}$$

The derivatives of $V_4$ with respect to time t is as follows:

$$\dot{V}_4 = \frac{1}{2}\dot{E}_k{}^T P_{k1} E_k + \frac{1}{2} E_k{}^T P_{k1}\dot{E}_k + \frac{1}{2}\dot{E}_{km}{}^T P_{k2} E_{km} + \frac{1}{2} E_{km}{}^T P_{k2}\dot{E}_{km} + \frac{1}{2\gamma_{kw}}\widetilde{W}_k{}^T\dot{\widetilde{W}}_k + \frac{1}{2\gamma_{ku}}\widetilde{U}_k{}^T\widetilde{U}_k \tag{68}$$

According to the same proof process as (65) and (66), $V_4 > 0$, $\dot{V}_4 \leq 0$. Thus, the above system is Lyapunov stable.

### 3.2.3. Schematic Diagram of the Control Algorithm

Based on the LESO constructed in this paper, this section uses the state variables estimated by LESO to design a state feedback robust controller. When constructing the state feedback robust controller, based on the construction of the feedback controller, the aggregate disturbance estimates $\hat{z}_{q5}$ and $\hat{z}_{k3}$ obtained by the LESO are used to compensate the unknown disturbance term. The adaptive RBFNN $W^T\phi(z)$ is used to approximate the state feedback gain matrix, while the adaptive RBFNN $U^T\varphi(\xi)$ is used for input saturation compensation (ISC). The robust controllers $v_{vs}$ and $v_{rc}$ are constructed to eliminate the influence of neural network approximation error and external disturbance, and improve the robust performance of the system. Finally, the feedback control input of the VSJ state-space model is obtained by the feedback linearization (FL) method. The control scheme of the designed FL+LESO+RBFNN+ISC controller is shown in the Figure 3.

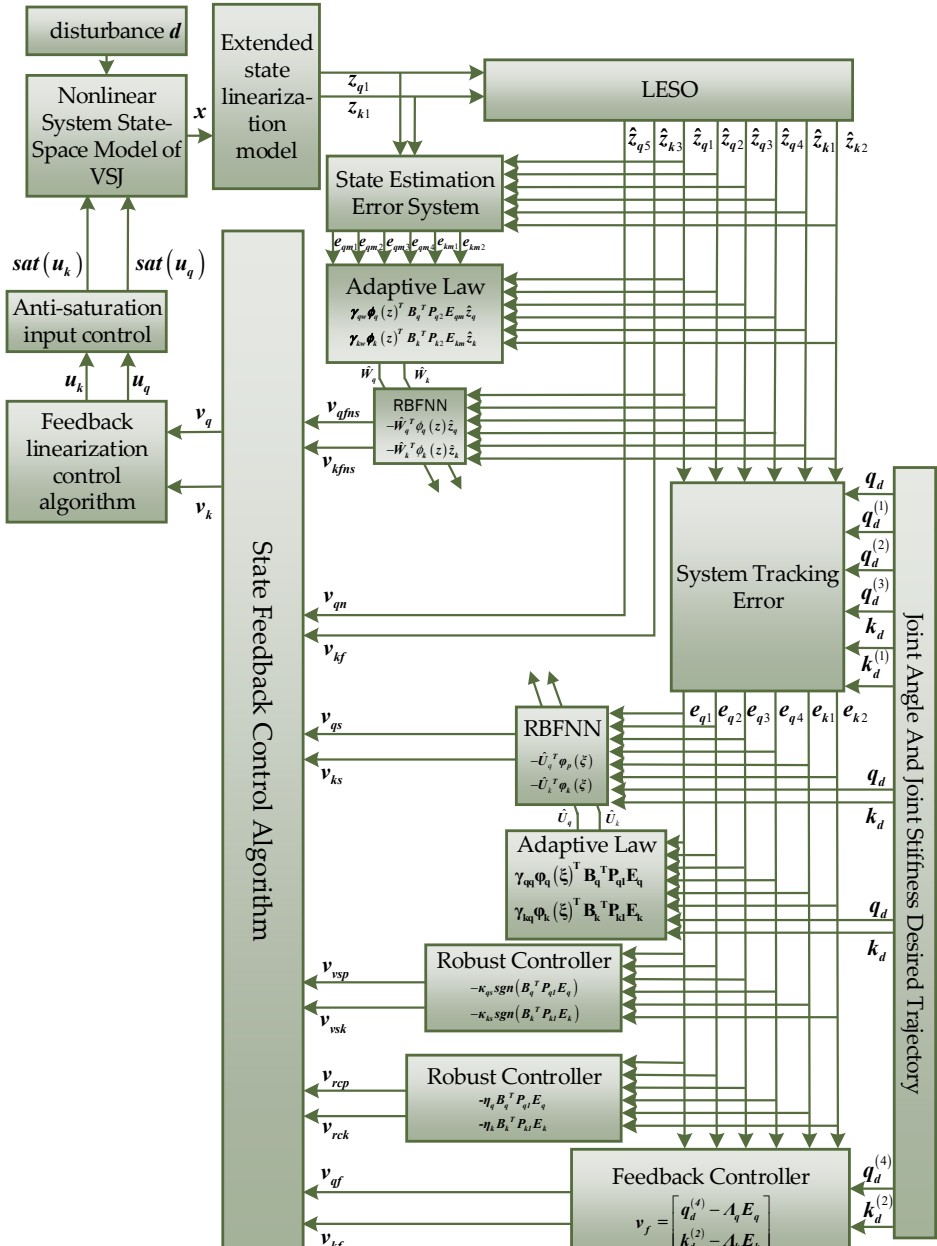

**Figure 3.** The scheme diagram of the FL+LESO+RBFNN+ISC controller. FL: feedback linearization; LESO: linear extended state observer; RBFNN: Radical Basis Function Neural Network; ISC: input saturation compensation.

## 4. Simulation

This section simulates the tracking performance of joint angle and joint stiffness of the VSJ on MATLAB-Simulink.

### 4.1. Simulation Setting

The space manipulator structure model designed by the modeling software NX in this paper is shown in Figure 4. From the model, the equivalent moment of inertia and equivalent friction damping coefficient of the VSJ can be obtained as shown in Table 1 by choosing the similar materials and structures as the experimental system. And other system parameters shown in Table 1 are setting according to the physical parameters of the hardwares of the experimental system below.

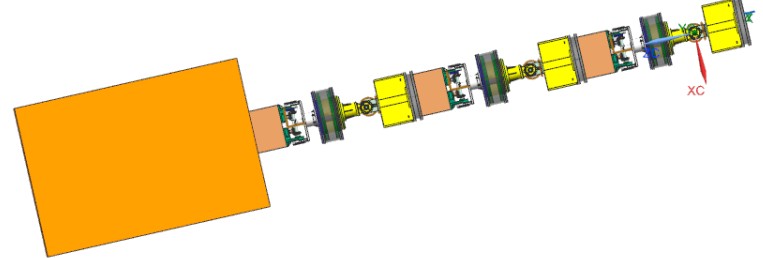

**Figure 4.** The space manipulator structure model.

**Table 1.** VSJ system parameters.

| Parameters | Value | Unit |
|---|---|---|
| VSJ Equivalent Moment of Inertia $H$ | 0.09 | Kg m$^2$ |
| VSJ Equivalent Friction Damping Coefficient $C$ | 0.2 | N m/rad |
| VSJ Angle Control Motor Equivalent Moment Of Inertia $J_p$ | 0.105 | Kg m$_2$ |
| VSJ Stiffness Control Motor Equivalent Moment Of Inertia $J_s$ | 0.00082 | Kg m$^2$ |
| VSJ Angle Control Motor Equivalent Damping Coefficient $C_p$ | 10.255 | N m/rad |
| VSJ Stiffness Control Motor Equivalent Damping Coefficient $C_s$ | 0.158 | N m/rad |
| Stiffness coefficient of the internal spring of the joint $k_s$ | 10,000 | N/m |
| Length of the inner lever chute of the joint $\Omega$ | 0.015 | m |
| Gear rack gear ratio n | 0.006 | m/rad |

The range of joint angle deviation of the VSJ is limited to $\phi = \pm 0.35 rad$.

After the VSJ-based space manipulator is in contact with the target, the joint angle can be oscillated and attenuated under controlled conditions until it is stable, and the joint stiffness also needs to be adjusted by oscillation. Therefore, the expected trajectory of the joint angle and joint stiffness of the VSJ designed in this paper are designed as follows:

$$
\begin{aligned}
q_d(t) &= 2 - 2\sin(6t)e^{-0.5t} \\
k_d(t) &= \left(20\sin(6t)e^{-0.5t} + 20\sin(6t)\right)sign\left(\sin(6t)e^{-0.5t} + \sin(6t) + 24.5\right)
\end{aligned}
\tag{69}
$$

The system state disturbances, parameter uncertainties and unknown bounded frictional torques, as well as unknown bounded external disturbances are shown in the Table 2.

**Table 2.** VSJ system compound disturbances.

| Parameters | Value | Parameters | Value |
|---|---|---|---|
| $\Delta H$ | $0.4 \times H$ | $\Delta C_s$ | $0.5 \times C_s$ |
| $\Delta C$ | $0.4 \times C$ | $\tau_{pf}$ | 0.3 N m |
| $\Delta J_p$ | $0.5 \times J_p$ | $\tau_{sf}$ | 0.2 N m |
| $\Delta J_s$ | $0.5 \times J_s$ | $\tau_e$ | 5 N m |
| $\Delta C_p$ | $0.5 \times C_p$ | | |

Comparing the algorithm of this paper with the FL-LESO-SMC-ISC algorithm in [37] by simulation. The control parameter settings of the two algorithms are shown in Table 3.

**Table 3.** VSJ system control algorithm parameters.

| Algorithm | Parameters | Value | Parameters | Value |
|---|---|---|---|---|
| FL-LESO-SMC-ISC | $L_e$ | $\begin{bmatrix} 200 & 16000 & 6.4\times10^5 & 1.28\times10^7 \\ 0 & 0 & 0 & 0 \\ 0 & 0 & 1.024\times10^8 & 0 \\ 120 & 4800 & 0 & 6.4\times10^4 \end{bmatrix}^T$ | $(\gamma_q,\gamma_k)$ | (150,150) |
| | $c_q$ | (2000,500,40) | $(p_q,p_k)$ | (1/15, 1/15) |
| | $c_k$ | 50 | $(\rho_q,\rho_k)$ | (50,30) |
| | $(r_q,r_k)$ | (5,5) | $sat(u_p)$ | [−40.0 N m, 40.0 N m] |
| | $(\mu_q,\mu_k)$ | (5,5) | $sat(u_k)$ | [−10.0 N m, 10.0 N m] |
| FL-LESO-RBFNN- ISC | $L_e$ | $\begin{bmatrix} 200 & 16000 & 6.4\times10^5 & 1.28\times10^7 \\ 0 & 0 & 0 & 0 \\ 0 & 0 & 1.024\times10^8 & 0 \\ 120 & 4800 & 0 & 6.4\times10^4 \end{bmatrix}^T$ | $b_{\varphi qj}$ | 2 |
| | $\Lambda_q$ | (10000,5000,800,50) | $b_{\varphi kj}$ | 20 |
| | $\Lambda_k$ | (100,20) | $(\gamma_{qu},\gamma_{qw})$ | (0.3, 0.35) |
| | $c_{\phi qj}$ | [−5,−4,−3,−2,−1,0,1,2,3,4,5] | $(\gamma_{ku},\gamma_{kw})$ | (0.3, 0.35) |
| | $c_{\phi kj}$ | [−40,−30,−20,−10,−0,10,20,30,40] | $(\kappa_{qs},\kappa_{ks})$ | (0.5, 0.5) |
| | $b_{\phi qj}$ | 2 | $(\eta_q,\eta_k)$ | (0.1,0.1) |
| | $b_{\phi kj}$ | 20 | $sat(u_p)$ | [−40.0 N m, 40.0 N m] |
| | $c_{\varphi qj}$ | [−5,−4,−3,−2,−1,0,1,2,3,4,5] | $sat(u_k)$ | [−10.0 N m, 10.0 N m] |
| | $c_{\varphi kj}$ | [−40,−30,−20,−10,−0,10,20,30,40] | | |

The positive definite symmetric matrix used by the FL-LESO-RBFNN-ISC algorithm is calculated by (26) and (50) as follows:

$$P_{q1} = \begin{bmatrix} 16848.0 & 8418.0 & 1168.4 & 0.0005 \\ 8418.0 & 4388.1 & 668.4047 & 1.6846 \\ 1168.4 & 668.4047 & 133.8713 & 1.6846 \\ 0.0005 & 1.6846 & 0.8418 & 0.1168 \end{bmatrix};$$

$$P_{q2} = \begin{bmatrix} 0.0341 & 1.8188 & 36.375 & 3.9062\times10^{-7} \\ 1.8188 & 872.875 & 2.9095\times10^4 & 4.364\times10^5 \\ 36.375 & 2.9095\times10^4 & 1.3096\times10^6 & 2.328\times10^7 \\ 3.9062\times10^{-7} & 4.364\times10^5 & 2.328\times10^7 & 4.656\times10^8 \end{bmatrix}$$

(70)

$$P_{k1} = \begin{bmatrix} 26.25 & 0.05 \\ 0.05 & 0.2525 \end{bmatrix}; P_{k2} = \begin{bmatrix} 0.0417 & 0.001 \\ 0.001 & 200.1667 \end{bmatrix}$$

(71)

$$Q_{q1} = \begin{bmatrix} 10 & 0 & 0 & 0 \\ 0 & 10 & 0 & 0 \\ 0 & 0 & 10 & 0 \\ 0 & 0 & 0 & 10 \end{bmatrix}; Q_{q2} = \begin{bmatrix} 10 & 0 & 0 & 0 \\ 0 & 10 & 0 & 0 \\ 0 & 0 & 10 & 0 \\ 0 & 0 & 0 & 10 \end{bmatrix}; Q_{k1} = \begin{bmatrix} 10 & 0 \\ 0 & 10 \end{bmatrix}; Q_{k2} = \begin{bmatrix} 10 & 0 \\ 0 & 10 \end{bmatrix}$$

(72)

*4.2. Simulation Results*

Figures 5–7 are the simulation results, where Figure 5 includes the VSJ angle and stiffness tracking output responses, the VSJ angle and stiffness tracking error VSJ angle tracking errors and the VSJ angle and stiffness tracking control torque. Figure 6 shows the length of VSJ lever arm, while Figure 7 shows VSJ angle deviation between $q$ and $\theta_p$.

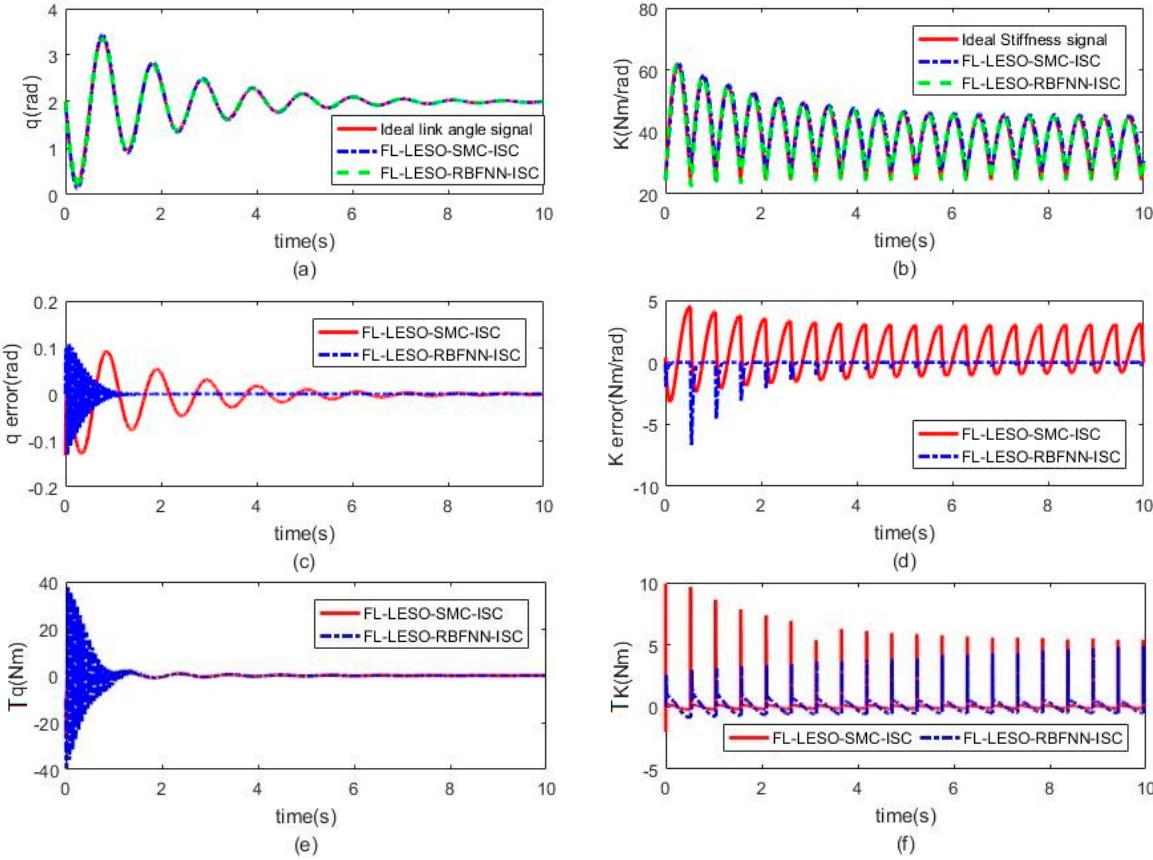

**Figure 5.** Simulation results. (**a**) VSJ angle tracking output response of two control algorithm. (**b**) VSJ stiffness tracking output response of two control algorithm. (**c**) VSJ angle tracking error VSJ angle tracking error of two control algorithm. (**d**) VSJ stiffness tracking error VSJ angle tracking error of two control algorithm. (**e**) VSJ angle tracking control torque of two control algorithm. (**f**) VSJ stiffness tracking control torque of two control algorithm.

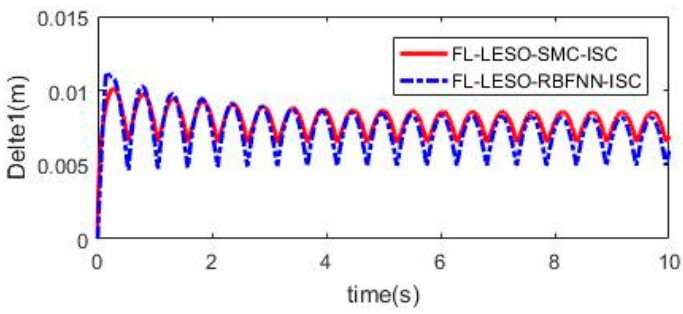

**Figure 6.** VSJ lever arm.

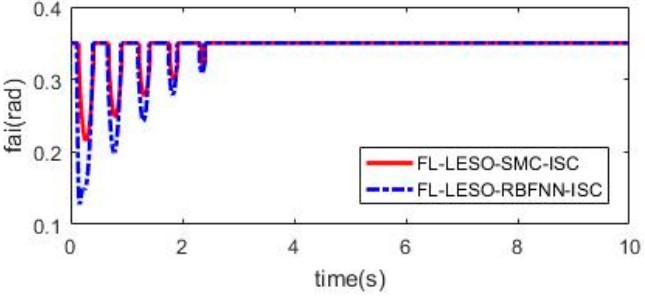

**Figure 7.** VSJ angle deviation.

## 5. Experiment

### 5.1. Experimental Setting

The semi-physical experiment system of the space robot VSJ single joint principle prototype is constructed with system parameters shown in Table 1, although there will be some deviation. The main hardware architecture includes: a computer for data processing, an FPGA control card as the core control module, and a space robot VSJ single joint prototype. The communication module of joint control system adopts CAN bus. The single joint prototype is placed on air floating platform supported by four air feet and has a yaw freedom. It is constructed mainly by the joint motor (Kollmorgen TBM(S)-12955-X, Radford, VA., USA) and the stiffness motor (Kollmorgen TBM(S)-12913-X, Radford, VA., USA). The six-dimensional force sensor (ATI-Nano17, ATI INDUSTRIAL AUTOMATION, Goodworth, NC., USA) is used for measuring the axial forces and moments. The encoder (EAC58P, ELCO Industrie Automation GmbH, Oberstenfeld, Bayern., Germany) is used for measuring the angular displacement and speed. The motor drive (HAR-5/60, Elmo Motion Control, Petah Tiqwa, Hamerkaz, Israel) is used for controlling motor motion. The diagram of the experimental system is shown in Figure 8. The workflow of the semi-physical experiment system is as follows: the operator operates on the PC and issues commands to the joint control system. The joint control system then sends the command to the motor drive, which rotates to change the joint angle and stiffness. At the same time, the joint control system uses the motor encoder to monitor the movement of the motor, and the position and speed information of the feedback motor are transmitted to the joint control module and the driver respectively, and the motor movement is adjusted according to the error, thereby adjusting the movement of the joint to achieve the purpose of movement and precision. Requirements to make the joints have better positioning. The experiment system uses a single pendulum to strike the end of the joint to produce an external disturbance force. In order to be able to generate the force in the yaw direction, the experiment was designed to strike the end of the joint at a certain angle to the pitch axis of the joint. The method of calculating the end impact force is as follows:

$$f_{yaw} = \text{F}\sin\theta = \frac{m \cdot (v_0 - v_t)}{\triangle t} \sin\theta \tag{73}$$

where F is the total collision force, $f_{yaw}$ is the collision force in the yaw direction, m is the mass of the pendulum block, $\triangle t$ is the collision duration, $v_0$ is the initial velocity of the pendulum block during the collision, and $v_t$ is the velocity of the pendulum block after the collision, and have:

$$v_0 = \sqrt{2gl(1 - \cos\alpha_0)}; v_t = \sqrt{2gl(1 - \cos\alpha_t)} \tag{74}$$

where g is the acceleration of gravity, $l$ is the distance from the fixed end of the inelastic light rope to the centroid of the pendulum block, $\alpha_0$ is the angle between the rope and the vertical direction at the starting position of the pendulum block, and $\alpha_t$ is the angle with the vertical direction when the pendulum block reaches the highest position after the collision. In this experiment, the parameters setting are as follows: $\theta = 30°$, $\alpha_0 = 10°$, $l = 0.2$ m , $g = 9.8$ m $\times$ s$^{-2}$, m $= 5$ kg, $t = 0.01$ s, the experimental measurements obtains that $\alpha_t \approx 3.8°$, so $F \approx 10$ N and $f_{yaw} \approx 5$ N.

At the same time, There are deviations between the actual values of the system parameters of the experiment system and the theoretical value in the kinetic equation of the system, which makes the system unknown uncertainty. This can be used to test the effectiveness of the algorithm.

The control algorithms used in the experiment system are written by MATLAB. The control algorithm parameter settings are shown in Table 3. The joint angle and stiffness curve tracked by the system are also shown in (75).

$$\begin{aligned} q_d(t) &= -2\sin(6t)e^{-0.5t} \\ k_d(t) &= \left(12\sin(6t)e^{-0.5t} + 12\sin(6t)\right)sign\left(\sin(6t)e^{-0.5t} + \sin(6t) + 24.5\right) \end{aligned} \tag{75}$$

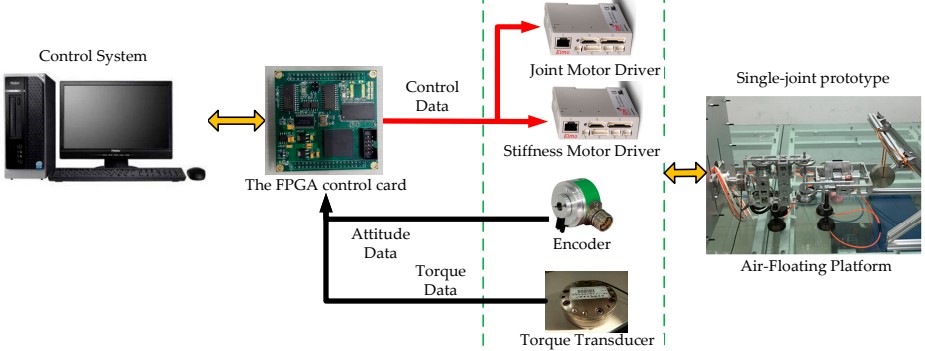

**Figure 8.** The diagram of the experimental system.

## 5.2. Experimental Results

Figures 9–15 are the experiment results, where Figure 9 includes the VSJ angle and stiffness tracking output responses, the VSJ angle and stiffness tracking error VSJ angle tracking errors and the VSJ angle and stiffness tracking control torque. Figure 10 shows the VSJ joint motor angle. Figure 11 shows the VSJ stiffness motor angle. Figure 12 shows the length of VSJ lever arm. Figure 13 shows the VSJ angle deviation between $q$ and $\theta_p$. Figure 14 shows the joint motor elastic torque and Figure 15 shows the stiffness motor resistant torque.

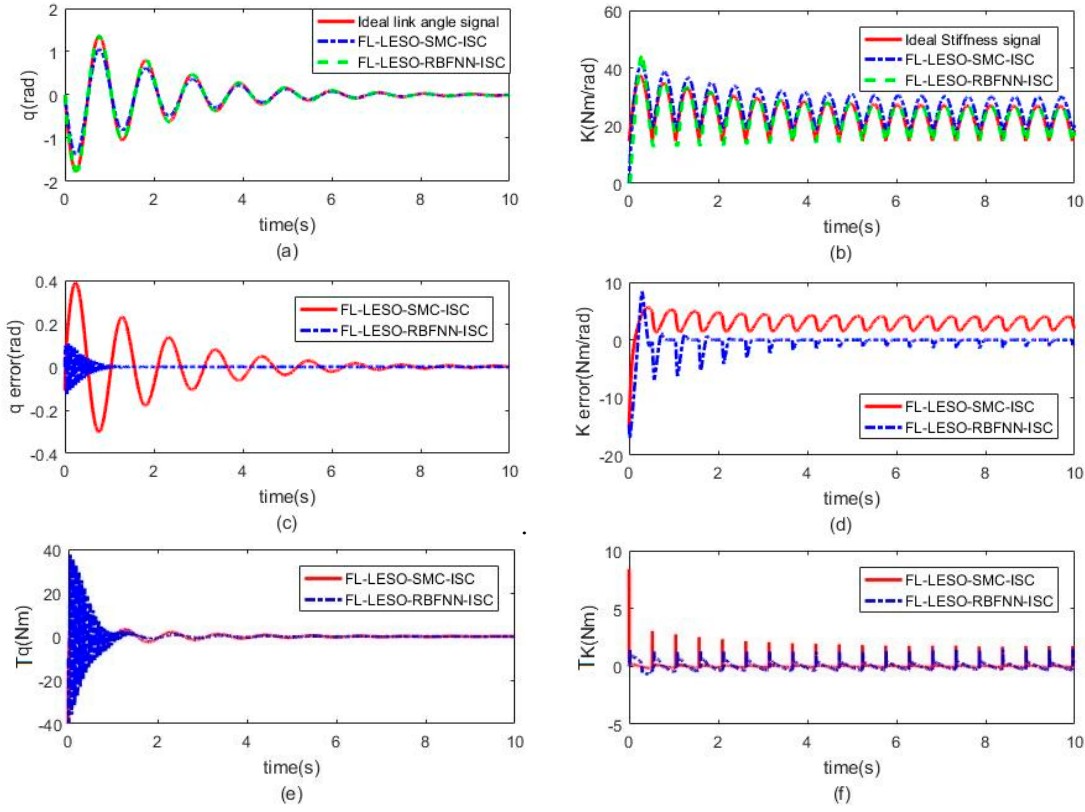

**Figure 9.** The experiment results. (**a**) VSJ angle tracking output response of two control algorithm. (**b**) VSJ stiffness tracking output response of two control algorithm. (**c**) VSJ angle tracking error VSJ angle tracking error of two control algorithm. (**d**) VSJ stiffness tracking error VSJ angle tracking error of two control algorithm. (**e**) VSJ angle tracking control torque of two control algorithm. (**f**) VSJ stiffness tracking control torque of two control algorithm.

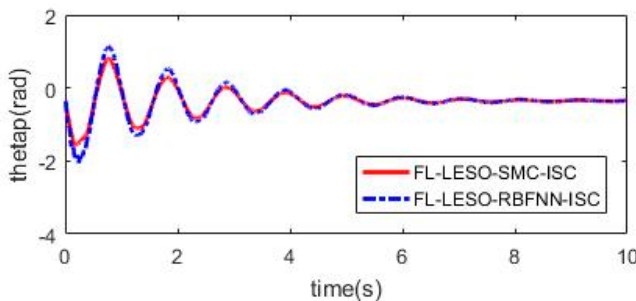

**Figure 10.** VSJ joint motor angle.

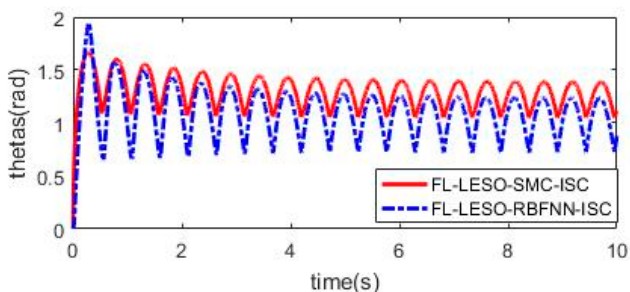

**Figure 11.** VSJ stiffness motor angle.

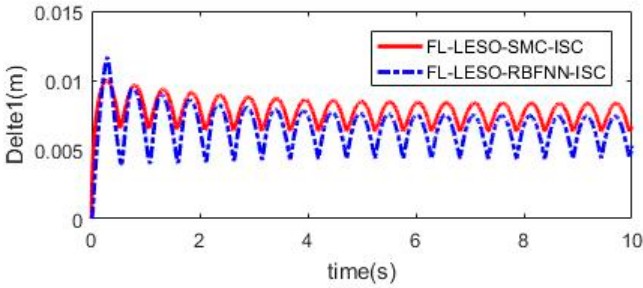

**Figure 12.** VSJ lever arm.

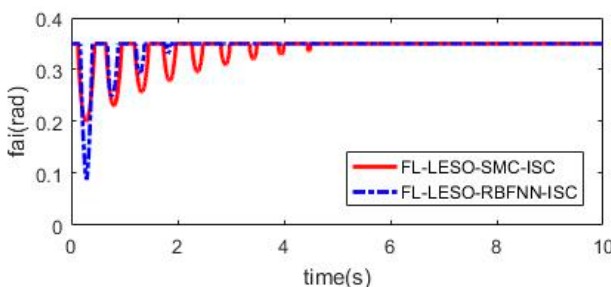

**Figure 13.** VSJ angle deviation.

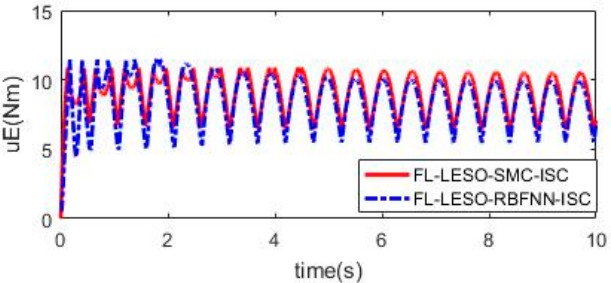

**Figure 14.** The joint motor elastic torque.

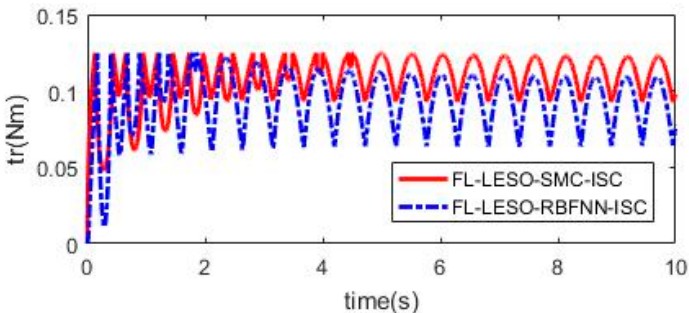

**Figure 15.** The stiffness motor resistant torque.

## 6. Discussion

In the previous section, the effectiveness of the proposed algorithm is verified by comparative simulation and semi-physical experiments. It can be seen from Figure 5a,b that both control algorithms can achieve coordinated tracking control of joint angle and joint stiffness during the simulation process. It can be seen from Figure 5c,d that the tracking error of the proposed algorithm is larger at the beginning of the simulation, but the convergence speed is much faster than the FL-LESO-SMC-ISC algorithm. This shows that the FL-LESO-RBFNN-ISC algorithm can better approximate the tracking object through the "universal approximation" feature of the RBFNN. It shows that the FL-LESO-RBFNN-ISC algorithm has better influence on unknown interference than the FL-LESO-SMC-ISC algorithm. Figure 5e,f are the control torque input response curves of the simulation system. It can be seen from the curve that the input torques of the two control algorithms are within the allowable range. At the initial moment after the end of the collision (i.e., near zero), the control inputs of the two algorithms reach a maximum. This indicates that the input saturation compensation measures in both algorithms are valid. Figures 6 and 7 show the length of the lever arm and the joint angle deviation in VSJ during the simulation, respectively. It can be seen from the curves in the figure that the changes are within the allowable range of the joint model used in this paper, and the simulation results are effective.

In the semi-physical experiment to verify the two control algorithms, the stiffness of the joint can be obtained by (2) based on the measured motor and joint attitude data. As can be seen from Figure 9a,b, both control algorithms can achieve coordinated tracking control of joint angle and joint stiffness during semi-physical testing. However, as can be seen from Figure 9c,d, in the initial stage of the experiment, since the initial reference value is different from the actual value, the tracking result has a "peak phenomenon". However, as the experiment progressed, the phenomenon gradually disappeared. The error of the entire experiment is slightly larger than the simulation error, which is caused by many unknown disturbances in the process of constructing the experiment system. From the comparison results of the error, the error convergence speed of the proposed algorithm is much faster than the FL-LESO-SMC-ISC tracking algorithm, and its tracking performance is better, especially in the case of spikes in the tracking curve. Figure 9e,f are the control torque input response curves of the experiment system. The curves show that, at the initial stage of the experiment, the joint angle control torque is saturated, and the anti-saturation and saturation compensation measures play a role. As the experiment progresses, the control torque gradually decreases. At the same time, the joint stiffness control torque is also close to saturation at the beginning and then gradually decreases. In addition, this experiment measured angular changes in the joint motor and the stiffness motor, as shown in Figures 10 and 11. Using the measured attitude parameters, the lever arm in Figure 12 and the angle deviation in Figure 13 can be obtained, and the values of both parameters are within the normal working range. Figures 14 and 15 show the calculated joint motor elastic torque and the stiffness motor resistant torque which is much smaller than the control torque during the experiment. All within the range of normal operation. The conclusion can be drawn from the experiment: the proposed algorithm has better trajectory tracking and anti-disturbances ability.

## 7. Conclusions

In order to solve the problem of controllable attenuation of the vibration of the flexible space manipulator after collision with the target, the VSJ is introduced to construct the flexible space manipulator and its dynamic model. Because its dynamic model is a complex nonlinear system, the original dynamic equation is transformed into a pseudo-linear system with integral chain type of input saturation constraint and matching lumped disturbance by means of coordinate transformation and feedback linearization. In order to realize the cooperative control of the angle and stiffness of the VSJ, this paper expands the matched lumped disturbance into a new system state and obtains an extended integral chain pseudo-linear system and constructs a LESO to estimate the unknown system state in the pseudo-linear system. Based on the idea of state feedback control, the neural state feedback controller, the neural network compensator for compensating unknown external time-varying interference and the neural network compensator with input saturation compensation are constructed by RBFNN. Then the stability analysis based on Lyapunov candidate function is used to prove the final uniform bounded stability of the constructed system. In this paper, the proposed algorithm is compared with the FL-LESO-SMC-ISC algorithm which uses the LESO for state estimation. The simulation and experiment on the semi-physical experiment platform prove the effectiveness and superiority of the proposed algorithm. The simulation and experimental results show that the proposed algorithm can track the target curve with higher precision. The algorithm proposed in this paper can better compensate the error when there is unknown disturbance. At the same time, the designed input saturation compensation controller can ensure that the output control torque is within its threshold range. The shortcoming of this paper is that for good position and stiffness tracking control, manual adjustment of controller parameters is required, and there is a trade-off between the tracking response performance of the system output and the response performance of the control input. Therefore parameter adaptive adjustment is needed. Then, since LESO is a high gain observer, a good estimate of the state of the unknown system is obtained. However, this can lead to an amplification effect of the state measurement noise and may affect the tracking performance of the controller or even the stability of the system. So estimation error compensation is required for this.

**Author Contributions:** Project administration, Z.-H.D.; Supervision, J.-C.H.; Writing—original draft, X.Y.

**Funding:** This research received no external funding.

**Acknowledgments:** Thanks A.P. Ming Chu, M.S. Xiaolong Ju and M.S. Fengquan Gao for they building the semi-physical experiment system.

**Conflicts of Interest:** The author declares no conflict of interest.

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
