# Peer review of "Research on Angle and Stiffness Cooperative Tracking Control of VSJ of Space Manipulator Based on LESO and NSFAR Control"

_electronics, doi:10.3390/electronics8080893_

Round 1

Reviewer 1 Report

The paper presents a tracking controller for a variable stiffness jointed space manipulator through linear extended state observer and neural State Feedback Robust Control.  Overall the paper is written well, and relevant literatures are presented and addressed well.  Even though the approach is based on well-established control algorithms and procedures, the application of the technique on the space manipulator with various unbounded variables might make the contribution of the study acceptable. However, there are few points need to be addressed well before final publication

Section 4, Simulation setup and procedures need to be justified. For example, what is the justification for selecting the trajectory of the joint angle and stiffness of the VSJ as shown in equation 69? The range of the joint angle. I think the justification is needed with respect to the intended application and the nature of the unpredicted trajectory of the system. In the experimental setting and verification part, a different trajectory is planned for the joint and stiffness (equation 75), were there any physical limitations or other reasons not to keep it the same as the simulation part (equation 69)?

Reviewer 2 Report

This paper describes the control design for a variable stiffness joint based on feedback linearisation, linear extended state observer and Radial Basis Function Neural Network. Bounded stability is shown with the use of the Lyapunov function. The controller is tested experimentally in simulation and with the use of a simulated-zero-gravity physical system. 

The title is way too long and becomes word soup. It should be possible to summarise better than this.

The paper is generally well written but I do recommend someone with a good grasp of English grammar checks over the whole paper for readability. 

The introduction and literature review is reasonable, but many references are just listed with little critical comparisons between methods. For example, RBFNN reference is mentioned but no other NN methods are compared here. 

There is a general problem in this paper with the readability of the equations. Please check delimits between terms, and that all variables are expained in the text, including the use of subscripts. This makes it very hard to verify the truth to your control design.

Figure 1 does not appear to add anything to the paper and can probably be removed. 

Many equations are very busy on the page, which makes them difficult to read. For example, eq. (1) has several different equations very close together which can make it confusing. 

The explaination of parameters for eq. (1) in lines 208~215 is extremely hard to read and identify what they are. It also appears that variables q, θp and K are not described. The use of Δ as a variable is a poor choice as it makes other uses of Δ ( as in ΔK) confusing.

In eq. (4) g is described as a function of x, although it appears to only contain constants. Bigger matrices with many terms like those in (4) and (6) should be comma delimited for clarity.

In (7) L is not defined, and subscripts are not clearly defined, making the whole thing very hard to follow. 

Line 260, I am not sure of the validity of the statement made here (det(G(x)) always non-zero), please make this clear.

I believe it would make more sense to put Assumption 2 before the description of the system. 

Please explain where the parameters in table 1 have been taken from. 

Tables 2 and 3 need delimiting in the middle for clarity, as it is repeated headings in two columns. 

Please clarify why φ is limited to ±0.35 rad.

Please explain where the values in (70 - 72) are taken from. 

Figure 5 should appear later in the text.

Please clarify what is being shown in fig 5 (e-f), as it is described as torque but labelled with an F, normally used for force. 

Line 489, it appears the wrong figure is being referenced.

Although a collision force is described, it is not clear when the disturbance is being applied. 

The reference signals defined in (75) and (69) are different, why is this?

Are figure 14 and 15 necessary at all for the discussion?
